# Compressing and Debiasing Vision-Language Pre-Trained Models for Visual Question Answering

**Qingyi Si**[1,2*]**, Yuanxin Liu**[3*]**, Zheng Lin**[1,2†]
**Peng Fu**[1]**, Yanan Cao**[1,2]**, Weiping Wang**[1]

[1]Institute of Information Engineering, Chinese Academy of Sciences, Beijing, China
[2]School of Cyber Security, University of Chinese Academy of Sciences, Beijing, China
[3]National Key Laboratory for Multimedia Information Processing,
School of Computer Science, Peking University

{siqingyi,linzheng,fupeng,caoyanan,wangweiping}@iie.ac.cn, liuyuanxin@stu.pku.edu.cn

## Abstract

Despite the excellent performance of vision-language pre-trained models (VLPs) on conventional VQA task, they still suffer from two problems: First, VLPs tend to rely on language biases in datasets and fail to generalize to out-of-distribution (OOD) data. Second, they are inefficient in terms of memory footprint and computation. Although promising progress has been made in both problems, most existing works tackle them independently. To facilitate the application of VLP to VQA tasks, it is imperative to jointly study VLP compression and OOD robustness, which, however, has not yet been explored. This paper investigates whether a VLP can be compressed and debiased simultaneously by searching sparse and robust subnetworks. To this end, we systematically study the design of a training and compression pipeline to search the subnetworks, as well as the assignment of sparsity to different modality-specific modules. Our experiments involve 3 VLPs, 2 compression methods, 4 training methods, 2 datasets and a range of sparsity levels. Our results show that there indeed exist sparse and robust subnetworks, which are competitive with the debiased full VLP and clearly outperform the debiasing SoTAs with fewer parameters on OOD datasets VQA-CP v2 and VQA-VS.[1]

## 1 Introduction

Visual Question Answering (VQA) (Antol et al., 2015) is an important task at the intersection of CV and NLP. In the last decade, deep neural networks have made promising progress in VQA. However, recent studies (Agrawal et al., 2016; Manjunatha et al., 2019) have found that VQA models are prone to dataset biases. As a result, they always suffer from sharp performance drops when faced with out-of-distribution (OOD) test datasets, whose answer distributions are different from the training set.

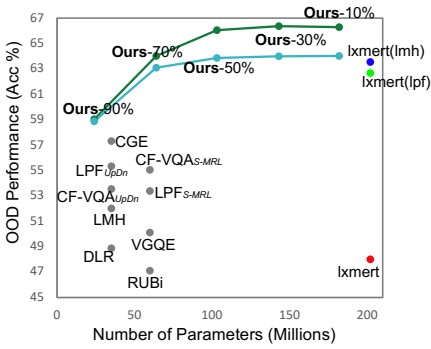

Figure 1: Comparison of accuracy and model sizes with debiasing SoTAs on VQA-CP v2. The green and cyan lines represent our "lxmert(lpf) + mask train(lmh)" and "lxmert(lmh) + mask train(lmh)", respectively, with modality-specific sparsity.

Although large-scale vision-language pre-trained models (VLPs) achieve further improvements in the in-distribution (ID) VQA benchmark (Goyal et al., 2017), they also fail to address the dataset-bias problem (Agrawal et al., 2018), e.g., lxmert (Tan and Bansal, 2019) suffers a 23.26% drop between ID and OOD accuracy. At the same time, the improvement brought by VLPs is partly due to their large model size, which increases the computational cost of deploying VQA models. To facilitate the application of VLPs to VQA tasks, the two problems should be addressed simultaneously. However, existing researches mostly focus on each of them separately.

The dataset-bias problem in VQA is well studied by numerous debiasing methods based on conventional small-scale models(Anderson et al., 2018; Cadene et al., 2019). Their main solution (Cadene et al., 2019; Clark et al., 2019; Liang et al., 2021b; Mahabadi and Henderson, 2019) is to regularize the loss according to the bias degree of training samples. In terms of the increased computational cost, a line of recent efforts have been made to compress pre-trained language models (PLMs) in the NLP

---
*Equal contribution. † Corresponding author: Zheng Lin.
[1]The codes can be found at https://github.com/PhoebusSi/Compress-Robust-VQA.

field (Chen et al., 2020b; Li et al., 2020a,b; Liang et al., 2021a; Liu et al., 2021, 2022; Prasanna et al., 2020) and VLPs for visual-linguistic tasks (Fang et al., 2021; Gan et al., 2022). They show that large-scale PLMs and VLPs can be compressed into lightweight models without degrading performance. Refer to **App. A** for more related work.

This paper jointly studies the compression and debiasing problems of VLP for the VQA task. To this end, we combine the existing debiasing and pruning methods to establish a training and compression pipeline, and conduct extensive experiments with the pre-trained lxmert, which is the most popular VLP in VQA, under different OOD settings. We show that there exist sparse lxmert subnetworks that are more robust than the full model, which suggests that the goal of OOD robustness and computational efficiency can be achieved simultaneously.

We also present a comprehensive study on the design of the training and compression pipeline, as well as the assignment of sparsity to different model modules, to identify subnetworks with better OOD generalization. Our findings highlight the importance of 1) Employing a two-stage training and compression pipeline and integrating the debiasing objective throughout the entire process. 2) If there are two debiasing methods working well with the full model, training the full model with the relatively poor-performing one and compressing it with the better one. 3) Assigning modality-specific sparsity to different modules of VLP.

Our main contributions are as follows: (1) We present the first (to our knowledge) systematic study on sparsity and OOD robustness for VLPs. (2) Our empirical studies on the training and compression pipeline and sparsity assignment can serve as a valuable guideline for the future design of VLP subnetwork searching methods. (3) We obtain subnetworks that outperform existing debiasing SoTAs in terms of the trade-off between accuracy and model size on OOD datasets VQA-CP v2 and VQA-VS (see Fig. 1, Tab. 1 and Tab. 2).

## 2 Method

### 2.1 VLP Architecture and Subnetworks

This section takes lxmert as an example to introduce how we extract subnetworks. Lxmert contains an embedding layer, a visual fc layer, a pooler layer, a VQA-specific classifier and a stack of Transformer layers, which involve three encoders: lan-

guage encoder ($L_{enc}$), object relationship encoder ($R_{enc}$) and cross-modality encoder ($C_{enc}$).

We adopt unstructured pruning to obtain a compressed version (i.e., a subnetwork) of the original VLPs. Specifically, given a VLP $f(\boldsymbol{\theta})$ with parameters $\boldsymbol{\theta}$, we apply a binary pruning mask $\mathbf{m} \in \{0, 1\}^{|\boldsymbol{\theta}|}$ to the model parameters, which gives rise to $f(\mathbf{m} \odot \boldsymbol{\theta})$, where $\odot$ is the element-wise product. The parameters to be pruned are:

$$\boldsymbol{\theta}_{pr} = \{\mathbf{W}_{\text{emb}}, \mathbf{W}_{\text{vis-fc}}, \mathbf{W}_{\text{plr}}\} \cup \boldsymbol{\theta}_{L_{enc}} \cup \boldsymbol{\theta}_{R_{enc}} \cup \boldsymbol{\theta}_{X_{enc}} \quad (1)$$

where $\mathbf{W}_{\text{emb}}$, $\mathbf{W}_{\text{vis-fc}}$ and $\mathbf{W}_{\text{plr}}$ are the weights of embedding layer, vision fc layer and pool layer, $\boldsymbol{\theta}_{L_{enc}} \cup \boldsymbol{\theta}_{R_{enc}} \cup \boldsymbol{\theta}_{X_{enc}}$ are the parameters of Transformer layers. More details of lxmert can be found in **App. B.1**. Another model visualBERT (Li et al., 2019), which is also used in our experiments, will be introduced in **App. B.2**.

### 2.2 Pruning Methods

We consider two representative pruning methods, i.e., magnitude-based pruning (Han et al., 2015) and mask training (Louizos et al., 2018; Ramanujan et al., 2020; Sanh et al., 2020; Sehwag et al., 2020).

**Magnitude-based Pruning** approximates the importance of model parameters based on their absolute values and eliminates the less important ones. We adopt the basic version of magnitude-based pruning, i.e., one-shot magnitude pruning (OMP). OMP can optionally be combined with further fine-tuning of the pruned subnetwork to recover the performance drop.

**Mask Training** directly optimizes the binary pruning mask $\mathbf{m}$ towards the given objectives. Specifically, each weight matrix $\mathbf{W} \in \mathbb{R}^{d_i \times d_o}$ is associated with two mask matrices, namely a binary mask $\mathbf{m} \in \{0, 1\}^{d_i \times d_o}$ and a real-valued mask $\hat{\mathbf{m}} \in \mathbb{R}^{d_i \times d_o}$. In the forward propagation, $\mathbf{m}$ is computed from $\hat{\mathbf{m}}$ through binarization:

$$\mathbf{m}_{i,j} = \begin{cases} 1 & \text{if } \hat{\mathbf{m}}_{i,j} \geq \phi \\ 0 & \text{else} \end{cases} \quad (2)$$

where $\phi$ is the threshold. Then, the original weight matrix $\mathbf{W}$ is replaced with a pruned one $\mathbf{m} \odot \mathbf{W}$. When it comes to backward propagation, we follow (Liu et al., 2022; Mallya et al., 2018; Radiya-Dixit and Wang, 2020; Zhao et al., 2020) and use the *straight-through estimator* (Bengio et al., 2013) to estimate the gradients of $\hat{\mathbf{m}}$ using the gradients of

m, and then update $\hat{\mathbf{m}}$ as $\hat{\mathbf{m}} \leftarrow \hat{\mathbf{m}} - \eta \frac{\partial \mathcal{L}}{\partial \mathbf{m}}$, where $\eta$ is the learning rate.

We initialize $\hat{\mathbf{m}}$ according to the magnitudes of the pre-trained weights of lxmert. This strategy is shown to be more effective than random initialization for pre-trained language models (Liu et al., 2022; Radiya-Dixit and Wang, 2020) and we also validate this in our experiments with lxmert (see **App. C.2**). Specifically, $\hat{\mathbf{m}}$ is initialized as:

$$\hat{\mathbf{m}}_{i,j} = \begin{cases} 0 & \text{if } \mathbf{W}_{i,j} \text{ is pruned by OMP} \\ \alpha \times \phi & \text{else} \end{cases} \quad (3)$$

where $\alpha \geq 1$ is a hyper-parameter. At initialization, we set the threshold $\phi = 0.01$ (any other value with the same order of magnitude should also be fine). To ensure that the subnetwork satisfies the given sparsity, $\phi$ is re-computed every $t_m$ training steps.

## 2.3 Debiasing Methods

The deabising methods in VQA usually contain a main model and a biased model. The biased model, which learns the language bias, is used to measure the training samples' bias degree and adjust the training loss for the main model. We experiment with SoTAs debiasing methods, i.e., LMH (Clark et al., 2019), RUBi (Cadene et al., 2019) and LPF (Liang et al., 2021b), of which LMH is widely studied for the OOD scenario of VQA (Chen et al., 2020a; Liang et al., 2020; Si et al., 2021) and NLU (Jia and Liang, 2017; McCoy et al., 2019; Schuster et al., 2019; Zhang et al., 2019). For comparison, we also describe the binary cross-entropy here.

**Binary Cross-Entropy** (BCE) computes the cross-entropy between the predicted distribution $\mathbf{p}_m$ (from main model) and the soft target score of each ground-truth $\mathbf{t}$, as:

$$\mathcal{L}_{bce} = t \cdot log(\delta(\mathbf{p}_m)) + (1-t) \cdot log(1 - \delta(\mathbf{p}_m))] \quad (4)$$

where $\delta$ denotes the sigmoid function.

**Learned-Mixin +H** (LMH) adds a biased model to learn biases during training, as follows:

$$\hat{\mathbf{p}}_{deb} = softmax(log(\mathbf{p}_m) + g(h)log(\mathbf{p}_b)) \\ g(h) = softplus(w \cdot h) \quad (5)$$

where $\mathbf{p}_b$ and $\mathbf{p}_m$ are the predicted distribution of biased model and main model, respectively. $g(h)$ determines how much to trust the learned biases, based on lxmert's last hidden representation $h$. Following (Clark et al., 2019), we directly use the answers' frequency under each question type as

$\mathbf{p}_b$[2]. To prevent $\mathbf{p}_b$ from being ignored, LMH also adds an entropy penalty item $R$ in the final loss:

$$\mathcal{L}_{lmh} = t \cdot log(\delta(\hat{\mathbf{p}}_{deb})) + (1-t) \cdot log(1 - \delta(\hat{\mathbf{p}}_{deb}))] + R \quad (6)$$

**RUBi** adopts a training strategy similar to LMH to regularize the main model's probability, and uses standard cross-entropy as the training loss:

$$\hat{\mathbf{p}}_{deb} = softmax(\mathbf{p}_m \cdot \delta(\mathbf{p}_b))$$
$$\mathcal{L}_{\text{rubi}} = -\frac{1}{N} \sum_k^N log(\hat{\mathbf{p}}_{deb}) [a_k] \quad (7)$$

**LPF** measures the bias degree as $\alpha_k = \mathbf{p}_b [a_k]$ to regularize the loss of the main model:

$$\mathcal{L}_{\text{lpf}} = \frac{-1}{N} \sum_k^N (1 - \alpha_k)^\gamma log(softmax(\mathbf{p}_m)) [a_k] \quad (8)$$

where the $\gamma$ is a tunable hype-parameter.

## 2.4 Problem Formulation

Given the pre-trained lxmert $f(\boldsymbol{\theta}_{pt})$, our goal is to find a subnetwork $f(\mathbf{m} \odot \boldsymbol{\theta}_{ft})$ that satisfies a target sparsity level $s$ and maximizes the OOD performance:

$$\max_{\mathbf{m}, \boldsymbol{\theta}_{ft}} (\mathcal{E}_{\text{OOD}}(f(\mathbf{m} \odot \boldsymbol{\theta}_{ft}))), \text{ s.t. } \frac{\|\mathbf{m}\|_0}{|\boldsymbol{\theta}_{pr}|} = (1-s) \quad (9)$$

where $\mathcal{E}_{\text{OOD}}$ denotes OOD evaluation, $\|\|_0$ is the $L_0$ norm and $|\boldsymbol{\theta}_{pr}|$ is the total number of parameters in $\boldsymbol{\theta}_{pr}$. This goal is achieved by searching the optimal $\mathbf{m}$ and $\boldsymbol{\theta}_{ft}$ in model training and compression.

Eq. 9 only specifies the overall sparsity. In this work, we also explore a finer-grained control over sparsity, which allocates different sparsity to different modules of lxmert, given that the overall sparsity is satisfied. Concretely, we consider three modules from different modalities, i.e., the language module, the visual module and the cross-modality module. The constraint in the optimization problem is then rewritten as[3]:

$$\text{s.t. } \frac{\|\mathbf{m}_{Lan}\|_0}{|\boldsymbol{\theta}_{Lan}|} = (1-s_L), \frac{\|\mathbf{m}_{Vis}\|_0}{|\boldsymbol{\theta}_{Vis}|} = (1-s_R), \frac{\|\mathbf{m}_X\|_0}{|\boldsymbol{\theta}_{X_{enc}}|} = (1-s_X),$$
$$s_L \cdot \frac{|\boldsymbol{\theta}_{Lan}|}{|\boldsymbol{\theta}_{pr}|} + s_R \cdot \frac{|\boldsymbol{\theta}_{Vis}|}{|\boldsymbol{\theta}_{pr}|} + s_X \cdot \frac{|\boldsymbol{\theta}_{X_{enc}}|}{|\boldsymbol{\theta}_{pr}|} = s \quad (10)$$

where $\boldsymbol{\theta}_{Lan} = \boldsymbol{\theta}_{L_{Enc}} \cup \{\mathbf{W}_{\text{emb}}\}, \boldsymbol{\theta}_{Vis} = \boldsymbol{\theta}_{R_{Enc}} \cup \{\mathbf{W}_{\text{vis-fc}}\}$ and $\boldsymbol{\theta}_{X_{Enc}}$ are model parameters of

---

[2] We use the same $\mathbf{p}_b$ in our implementation of LMH, RUBi and LPF. More details of LMH can be found in **App. B.3**

[3] For simplicity, the pooler layer's parameters(0.5M) are not included in eq. 10. We directly set it to the target sparsity $s$.

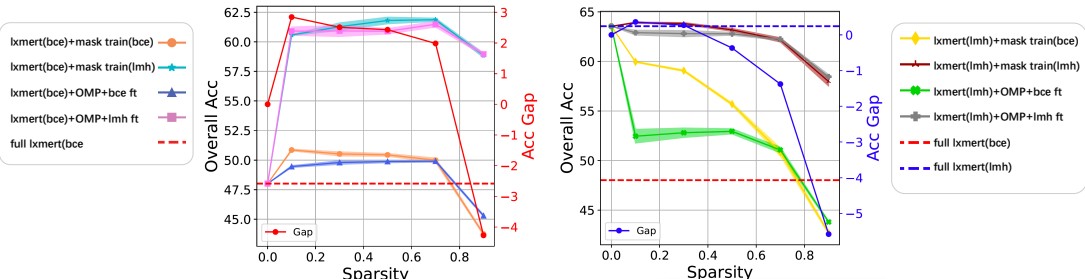

Figure 2: Results of subnetworks from the BCE fine-tuned lxmert (left) and from the LMH fine-tuned lxmert (right) on VQA-CP v2. "lxmert(bce/lmh)" denotes full model fine-tuning in Stage1, "mask train(bce/lmh)" and "OMP" denote pruning in Stage2. "bce/lmh ft" denotes further fine-tuning in Stage3. "Gap" denotes the improvement of mask train(bce/lmh) over full lxmert(bce/lmh). The shadowed areas denote standard deviations. These abbreviations are used throughout this paper. Detailed performance on three question types is shown in **App. C.1**

the language module, visual module and cross-modality encoder, respectively. $\mathbf{m}_{Lan}$, $\mathbf{m}_{Vis}$ and $\mathbf{m}_X$ are the binary masks for the three modules, respectively. $s_L$, $s_R$ and $s_X$ are the target sparsity levels for the three modules, respectively.

If not otherwise specified, we set the sparsity of every weight matrix to target sparsity. For example, if $s = 70\%$ and there is no modality-specific constraint, then all weight matrices are at 70% (**uniform sparsity**). If $s_L = 50\%$, then all weight matrices in $\boldsymbol{\theta}_{Lan}$ are at 50% sparsity, while $s_R$ and $s_X$ could be different (**modality-specific sparsity**).

### 2.5 Training and Compression Pipeline

We define two notations: $\mathcal{F}_{\mathcal{L}}(f(\boldsymbol{\theta}))$ denotes training $f(\boldsymbol{\theta})$ using loss $\mathcal{L} \in \{\mathcal{L}_{bce}, \mathcal{L}_{lmh}\}$. $\mathcal{P}_{\mathcal{L}}^p(f(\boldsymbol{\theta}))$ denotes pruning $f(\boldsymbol{\theta})$ using method $p \in \{\text{OMP, mask train}\}$ and loss $\mathcal{L}$ (if applicable), which outputs a pruning mask $\mathbf{m}$. A typical training and compression pipeline involves three stages:

**Stage1: Full Model Fine-tuning.** The pre-trained lxmert $f(\boldsymbol{\theta}_{pt})$ is fine-tuned using loss $\mathcal{L}$, which produces $f(\boldsymbol{\theta}_{ft}) = \mathcal{F}_{\mathcal{L}}(f(\boldsymbol{\theta}))$.

**Stage2: Model Compression.** The fine-tuned lxmert $f(\boldsymbol{\theta}_{ft})$ is compressed and we get the subnetwork $f(\mathbf{m} \odot \boldsymbol{\theta}_{ft})$, where $\mathbf{m} = \mathcal{P}_{\mathcal{L}}^p(f(\boldsymbol{\theta}_{ft}))$.

**Stage3: Further Fine-tuning (optional).** The subnetwork $f(\mathbf{m} \odot \boldsymbol{\theta}_{ft})$ is further fine-tuned using loss $\mathcal{L}'$, and gets $f(\mathbf{m} \odot \boldsymbol{\theta}'_{ft}) = \mathcal{F}_{\mathcal{L}'}(f(\mathbf{m} \odot \boldsymbol{\theta}_{ft}))$.

## 3 Experiments

In this section, we mainly investigate three questions: **(1)** How does compression affect lxmert's OOD generalization ability? **(2)** How to design the training and pruning pipeline to achieve a good sparsity-performance trade-off? **(3)** How to assign sparsity to different modality-specific modules?

### 3.1 Datasets, Model and Implementation

We conduct experiments on the OOD benchmarks VQA-CP v2 (Agrawal et al., 2018) and VQA-VS (Si et al., 2022b) that evaluate the robustness of VQA systems, with the accuracy-based evaluation metric (Antol et al., 2015). A more detailed discussion of the difference between the two datasets is shown in Sec. 3.5. We thoroughly study the above three questions on VQA-CP-v2, which is widely used in the literature on debiasing VQA systems (refer to Sec. 3.2, 3.3 and 3.4 ). Then, based on the findings, we further explore the more challenging VQA-VS (Si et al., 2022b) (refer to Sec. 3.5 ). For VLP, we adopt the lxmert-base-uncased model (Tan and Bansal, 2019) released by huggingface (Wolf et al., 2020). All the results are averaged over 4 random seeds. More information of the model and implementation details are shown in **App. B.4**.

### 3.2 Effect of Compression on OOD Accuracy

**Subnetworks from BCE Fine-tuned lxmert.** We compress the BCE fine-tuned lxmert using OMP and mask training and introduce either $\mathcal{L}_{bce}$ or $\mathcal{L}_{lmh}$ in the pruning (for mask training) or further fine-tuning process (for OMP).

The results are shown in the upper row of Fig. 2. We can derive several observations: 1) When no debiasing methods are used, the subnetworks of "mask train(bce)" and "OMP + bce ft" improve over the full lxmert by $1.35\% \sim 2.79\%$, even at up to 70% sparsity. This implies that lxmert is overparameterized and pruning may remove some parameters related to the bias features. 2) "mask train(lmh)" and "OMP + lmh ft" achieve further performance boost, exceeding full lxmert by a large margin ($11.05\% \sim 14.02\%$). Since mask training does not change the value of parameters, the

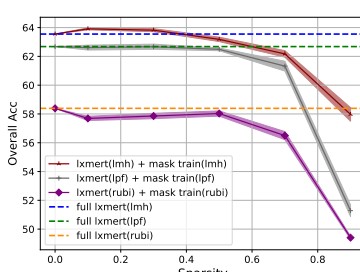

Figure 3: Results of lxmert subnetworks fine-tuned with different debiasing methods on VQA-CP v2.

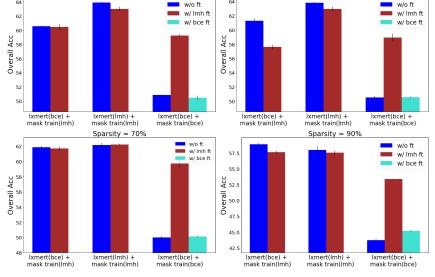

Figure 4: Results of lxmert subnetworks obtained from different training and compressing pipelines on VQA-CP v2. "ft" means further fine-tuning the subnetworks in **Stage3**.

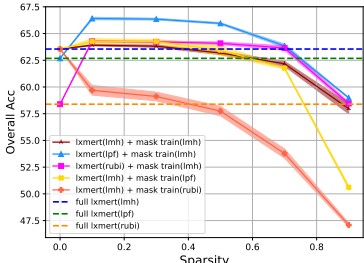

Figure 5: Results of lxmert subnetworks that adopt different debiasing methods in **Stage1** and **Stage2** on VQA-CP v2.

results of "mask train (lmh)" implicate that the biased "full lxmert(bce)" already contains sparse and robust subnetworks (across 10% ∼ 90% sparsity). 3) "mask train" outperforms "OMP" in general, which suggests that directly optimizing the subnetwork structure is more effective than debiasing a compressed subnetwork by further fine-tuning.

**Subnetworks from lxmert Fine-tuned with Debiasing Methods.** From the lower row of Fig. 2, we can find that: 1) For the full lxmert, the OOD performance is obviously promoted with the LMH debiasing method. 2) Unlike lxmert(bce) subnetworks, lxmert(lmh) subnetworks do not exhibit significant improvement over the full model. However, the "mask train(lmh)" and "OMP + lmh ft" subnetworks, which preserve the lxmert(lmh)'s performance at up to 50% sparsity, can serve as an efficient alternative to the LMH fine-tuned full lxmert. 3) "mask train(bce)" and "OMP + bce ft" clearly underperform their lmh counterparts, which suggests that it is important to use the debiasing method in pruning and subnetwork further fine-tuning even when the full model is already trained with the debiasing method.

Fig. 3 compares the subnetworks fine-tuned with LMH, LPF and RUBi. We find that: The subnetworks found using LMH consistently outperform those found by LPF and RUBi across different sparsity levels. Therefore, to save computing resources, we mainly use the best performing LMH in the following experiments and analysis.

### 3.3 Training and Compression Pipeline

In this section, we study the proper design of the training and compression pipeline, under the basic framework described in Sec. 2.5. Here we focus on the mask training compression method, as it

has been shown to generally outperform OMP with further fine-tuning. Our main observations can be described from three perspectives:

First, **it is recommended to introduce the debiasing loss across Stage1, Stage2 and (if applicable) Stage3.** The reason is three-fold: 1) As shown by Fig. 4, the subnetworks at 10%, 30% and 70% sparsity levels have better performance when starting from lxmert(lmh), as compared with the lxmert(bce). At 90% sparsity, "lxmert(lmh) + mask train(lmh)" underperforms "lxmert(bce) + mask train(lmh)" (see **App. C.3** for reasons), but the Accuracy gap is small. Therefore, adopting $\mathcal{L}_{lmh}$ in **Stage1** is a better choice than $\mathcal{L}_{bce}$, especially when the subnetworks are not at extremely high sparsity. 2) As we discussed in the previous section, introducing $\mathcal{L}_{lmh}$ in the mask training process (**Stage2**) substantially outperforms $\mathcal{L}_{bce}$ for both lxmert(lmh) and lxmert(bce). 3) When both Stage1 and Stage2 adopt the BCE loss, further fine-tuning the subnetworks with LMH loss in **Stage3** can significantly boost the performance, as shown by the results of "lxmert(bce) + mask train(bce)" w/o ft and w/ lmh ft in Fig. 4.

Second, **Stage3 is unnecessary if it adopts the same training objective as Stage2.** Comparing the blue and red (or cyan) bars in Fig. 4, we can see that further fine-tuning with the same training objective generally degrades the performance of "lxmert(lmh) + mask train(lmh)", "lxmert(bce) + mask train(lmh)" and "lxmert(bce) + mask train(bce)". This phenomenon suggests that **Stage3** can be eliminated to save computation cost.

Third, **it is recommended to use different debiasing methods in the two stages and leave the better one to Stage2.** As shown in Fig. 5, although LPF and RUBi are less effective in debi-

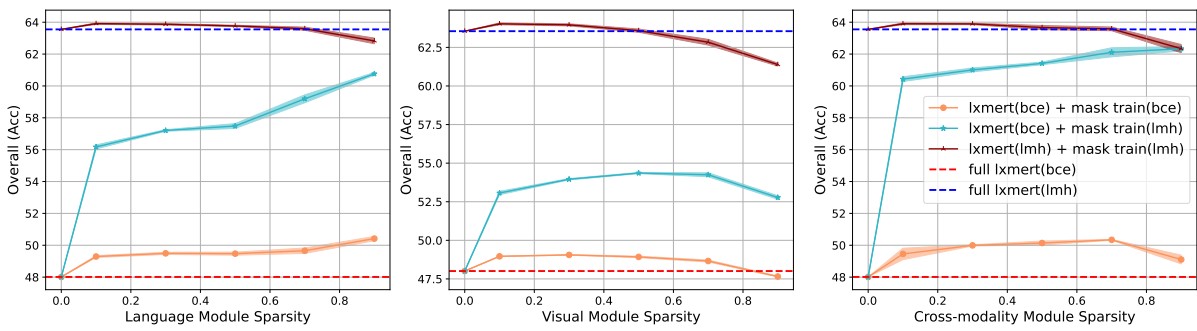

Figure 6: Results of subnetworks obtained by pruning the language (left), visual (middle) and cross-modality (right) modules. When pruning one module, the other two modules remain unpruned.

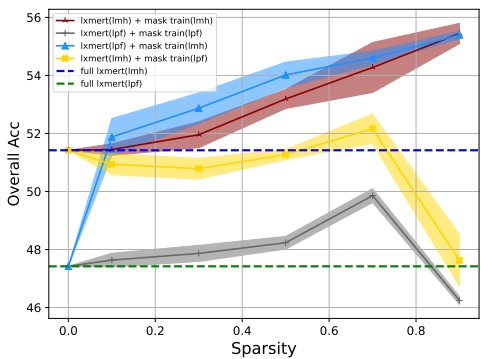

Figure 7: Results of visualBERT subnetworks that adopt different debiasing methods in **Stage1** and **Stage2** on VQA-CP v2.

asing the full model than LMH, "lpf+lmh"[4] and "rubi+lmh" are superior to "lmh+lmh". In contrast, when reversing the debiasing methods used in the two stages, "lmh+rubi" and "lmh+lpf" exhibit worse performance, suggesting that the better debiasing method should be used in **Stage2**. Additionally, "lpf+lmh" is superior to "rubi+lmh", which indicates that using a better debiasing objective in **Stage1** is helpful when we have multiple choices different from the **Stage2** objective. We also experiment with another VLP model, visualBERT (Li et al., 2019), and find that "lpf+lmh" still performs the best as in Fig. 7.

### 3.4 Modality-specific Sparsity

**Pruning Each Single Modality-specific Module.** Since lxmert uses different modules to encode the multi-modal data, it is intuitive to hypothesize that different modules of lxmert may capture the language bias to different extents. To validate this hypothesis, we compress the language, visual and cross-modality modules, respectively. As presented

[4] "lpf+lmh" denotes "lxmert(lpf) + mask train(lmh)"

by Fig. 6, the compression of different modality-specific modules indeed exhibits different effects.

When the full model is lxmert(bce) (the orange and cyan lines), compressing the language or cross-modality module has a positive effect on the OOD performance, and the accuracy generally improves as sparsity increases from 10% to 90%. By contrast, compressing the visual module results in inferior results than compressing the other two modules, even if the number of remaining parameters is larger (note that the visual module has a smaller number of parameters than the other two modules). These results suggest that, for the biased lxmert(bce), the language and cross-modality modules capture more training set bias than the visual module, which supports the above hypothesis.

In terms of "lxmert(lmh) + mask train(lmh)" (the red line), although compression does not lead to performance improvement like compressing lxmert(bce), the results also demonstrate that the language and cross-modality modules are more compressible than the visual module.

**Searching for Appropriate Modality-specific Sparsity.** Motivated by the above findings, we search for appropriate modality-specific sparsity by performing mask training with a variety of sparsity configurations (see **App. C.4**) for the three modules while keeping the overall sparsity the same.

As we can see in Fig. 8, at 50% and 70% overall sparsity, the configuration that achieves the best result assigns slightly higher sparsity to the language and cross-modality modules and significantly lower sparsity to the visual module, as compared with uniform sparsity. This phenomenon is in accordance with the findings in Fig. 6, implicating that compressing the three modules uniformly is suboptimal (at 50% ∼ 70% sparsity) and the language and cross-modality modules should be compressed to

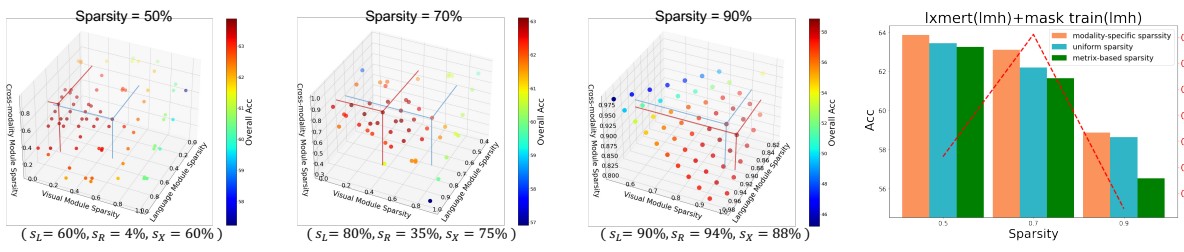

Figure 8: Results of subnetworks pruned by different sparsity configurations on VQA-CP v2 using "lxmert(lmh) + mask train(lmh)". Red and blue lines denote the coordinates of the data point with uniform sparsity across three modules and the data point performing the best (the specific configuration is shown below each plot) respectively. The overall sparsities are shown in the titles.

Figure 9: Comparison of different sparsity assignments on VQA-CP v2. "Gap" is the gap between "uniform sparsity" and "modality-specific sparsity".

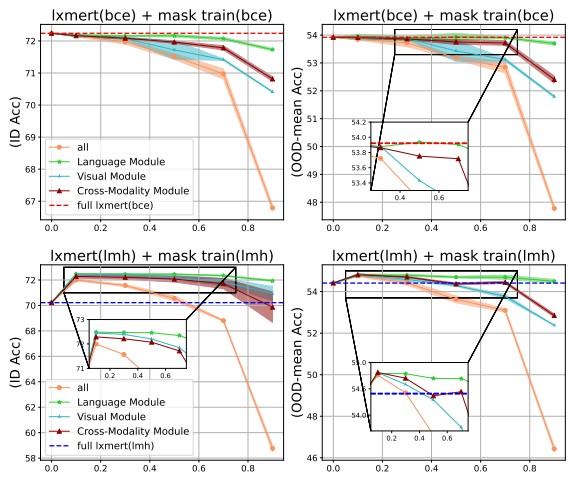

Figure 10: Results of subnetworks pruned using BCE (upper) and LMH (lower) on VQA-VS. We report ID and OOD-mean accuracy here. Results on specific OOD test sets are deferred to **App. D.1** Different lines denote subnetworks obtained by pruning all, language, visual and cross-modality modules respectively.

a larger extent than the visual module. At 90% sparsity, the sparsity configuration's comfort zone is in the proximity of the uniform point. Further increasing the sparsity of the language and cross-modality modules result in performance decline or only minor improvements. This is because 90% sparsity already approaches the compression upper bound, even for the language and cross-modality modules.

Fig. 9 shows a more direct comparison between the uniform and modality-specific sparsity. We also introduce another baseline "matrix-specific sparsity", which ranks all the model parameters, instead of the parameters in each weight matrix. This also results in different sparsity levels for different weight matrices, while there is no explicit control over the modality-specific sparsity. We can see that

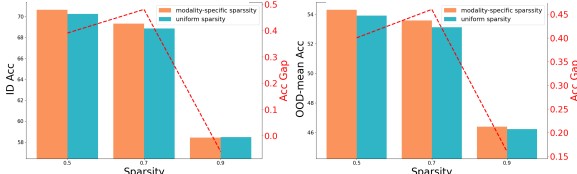

Figure 11: Comparison of "lxmert(lmh) + mask train(lmh)" subnetworks with uniform and modality-specific sparsity on VQA-VS. Results on specific OOD test sets can be found in **App. D.2**

modality-specific sparsity achieves the best results across the three overall sparsity levels from 50% to 90%, demonstrating its superiority. Besides, the results also suggest that, although simply allowing different matrices to have different sparsity is more flexible than uniform sparsity, it is not conducive to the final performance.

## 3.5 Exploration on VQA-VS

VQA-CP v2 is widely used in the literature of debiasing VQA systems. However, it only considers the question-type-based bias. To account for other potential biases, VQA-VS constructs several types of OOD test sets according to different shortcuts (e.g., keyword and key object). As a result, VQA-VS is more challenging and allows us to analyze the results on different biases. In this section, we search sparse and robust lxmert subnetworks in VQA-VS based on the major findings obtained from VQA-CP v2.

**The Effect of Compression.** Fig. 10 shows the results of full lxmert and subnetworks on VQA-VS. We can see that: 1) When using the BCE objective, we can identify sparse "bce+bce" subnetworks that are comparable with full lxmert (bce). 2) Different from VQA-CP v2, full lxmert (lmh) only slightly

| Methods | Backbone | Params. | All | Y/N | Num | Other |
|---|---|---|---|---|---|---|
| RUBi (Cadene et al., 2019) | S-MRL | ~60M | 47.11 | 68.65 | 20.28 | 43.18 |
| VGQE (Kv and Mittal, 2020) | S-MRL | ~60M | 50.11 | 66.35 | 27.08 | 46.77 |
| LPF (Liang et al., 2021b) | S-MRL | ~60M | 53.38 | 88.06 | 25.00 | 42.99 |
| CF-VQA (Niu et al., 2021) | S-MRL | ~60M | 55.05 | 90.61 | 21.50 | 45.61 |
| AdvReg. (Ramakrishnan et al., 2018) | UpDn | 35M | 41.17 | 65.49 | 15.48 | 35.48 |
| GRL (Grand and Belinkov, 2019) | UpDn | 35M | 42.33 | 59.74 | 14.78 | 40.76 |
| RUBi (Cadene et al., 2019) | UpDn | 35M | 44.23 | 67.05 | 17.48 | 39.61 |
| Loss-Rescaling (Guo et al., 2021) | UpDn | 35M | 47.09 | 68.42 | 21.71 | 42.88 |
| VGQE (Kv and Mittal, 2020) | UpDn | 35M | 48.75 | - | - | - |
| DLR (Jing et al., 2020) | UpDn | 35M | 48.87 | 70.99 | 18.72 | 45.57 |
| LMH (Clark et al., 2019) | UpDn | 35M | 52.01 | 72.58 | 31.12 | 46.97 |
| CF-VQA (Niu et al., 2021) | UpDn | 35M | 53.55 | 91.15 | 13.03 | 44.97 |
| LPF (Liang et al., 2021b) | UpDn | 35M | 55.34 | 88.61 | 23.78 | 46.57 |
| LMH+MMBS (Si et al., 2022a) | UpDn | 35M | 56.44 | 76.00 | 43.77 | 49.67 |
| CGE (Han et al., 2021) | UpDn | 35M | 57.32 | 87.04 | 27.75 | 49.59 |
| BCE | full lxmert | 202M | 48.01 | 48.24 | 20.04 | 55.57 |
| LPF (Clark et al., 2019) | full lxmert | 202M | 57.87 | 51.98 | | 52.58 |
| LMH (Clark et al., 2019) | full lxmert | 202M | 63.55 | 81.84 | 55.00 | 56.32 |
| **lpf+lmh (Ours)** | **10% lxmert** | **24M** | **59.05** | **75.08** | **57.12** | **51.17** |
| **lpf+lmh (Ours)** | **30% lxmert** | **64M** | **64.02** | **79.99** | **63.38** | **56.35** |
| **lpf+lmh (Ours)** | **50% lxmert** | **103M** | **66.07** | **84.70** | **63.71** | **56.95** |
| CE | full mPLUG | 350M | 57.05 | - | - | - |
| LPF (Clark et al., 2019) | full mPLUG | 350M | 65.24 | - | - | - |
| **ce+lpf (Ours)** | ~50% mPLUG | 182M | **62.53** | - | - | - |
| **lpf+lpf (Ours)** | ~50% mPLUG | 182M | **63.66** | - | - | - |

Table 1: Comparison with debiasing SoTAs on VQA-CP v2. "lpf+lmh" denotes "lxmert(lpf) + mask train(lmh)" subnetworks with modality-specific sparsity. "10% lxmert" denotes keeping 10% parameters of lxmert. The subnetworks from mPLUG are pruned using uniform sparsity.

| Methods | Backbone | Params. | ID | OOD-mean |
|---|---|---|---|---|
| Cross Entropy | S-MRL | ~60M | 62.03 | 42.65 |
| RUBi (Cadene et al., 2019) | S-MRL | ~60M | 59.09 | 38.73 |
| Cross Entropy | UpDn | 35M | 65.20 | 46.80 |
| LPF (Liang et al., 2021b) | UpDn | 35M | 54.72 | 43.31 |
| LMH (Clark et al., 2019) | UpDn | 35M | 56.89 | 45.85 |
| BCE | full lxmert | 202M | 72.24 | 53.92 |
| RUBi (Cadene et al., 2019) | full lxmert | 202M | 69.49 | 50.07 |
| LPF (Liang et al., 2021b) | full lxmert | 202M | 68.48 | 50.83 |
| LMH (Clark et al., 2019) | full lxmert | 202M | 70.22 | 54.41 |
| **bce+bce (Ours)** | **10% lxmert** | **24M** | **67.28** | **48.77** |
| **bce+bce (Ours)** | **30% lxmert** | **64M** | **70.89** | **53.06** |
| **bce+bce (Ours)** | **50% lxmert** | **103M** | **71.33** | **53.42** |
| **bce+bce (Ours)** | **70% lxmert** | **143M** | **71.85** | **53.51** |
| **bce+bce (Ours)** | **90% lxmert** | **183M** | **71.85** | **53.87** |
| **lmh+lmh (Ours)** | **10% lxmert** | **24M** | **58.42** | **46.39** |
| **lmh+lmh (Ours)** | **30% lxmert** | **64M** | **69.34** | **53.59** |
| **lmh+lmh (Ours)** | **50% lxmert** | **103M** | **70.66** | **54.31** |
| **lmh+lmh (Ours)** | **70% lxmert** | **143M** | **71.56** | **54.34** |
| **lmh+lmh (Ours)** | **90% lxmert** | **183M** | **71.97** | **54.75** |

Table 2: Comparison with debiasing SoTAs on VQA-VS. The subnetworks are pruned using modality-specific sparsity. "bce+bce" and "lmh+lmh" are defined in the same way as Tab. 1.

outperforms full lxmert (bce) in the OOD setting of VQA-VS, and underperforms in the ID setting. 3) The "lmh+lmh"[5] subnetworks improve over full lxmert (lmh) on both ID and OOD test sets, across a wide range of sparsity levels, suggesting that lxmert can also be simultaneously compressed and debiased on VQA-VS.

**The Effect of Modality-specific Sparsity.** Fig. 10 also shows that compressing different modality-specific modules has different effect on VQA-VS, as in VQA-CP v2. The language module is the most compressible while compressing the visual module results in the sharpest performance decline. To compare modality-specific sparsity and uniform sparsity, we directly inherit the sparsity configuration selected in Sec. 3.4 on VQA-CP v2. Fig. 11 shows that modality-specific sparsity consistently outperform uniform sparsity, except for 90% sparsity in the ID setting.

## 3.6 Comparison with Debiasing SoTAs

In this section, we will compare the best training and compression solutions identified in the previous sections with the current SoTA debiasing methods.

Tab. 1 shows the results on VQA-CP v2. We find that: 1) The accuracy of our methods (10% lxmert and 30% lxmert) beats the previous non-VLP debi-

---

[5]Since most debiasing methods (e.g., LPF and RUBi) fail on VQA-VS (see Tab.2), we only use LMH in VQA-VS. However, combining LMH and other effective debiasing methods in different stages may further outperform "lmh+lmh", as found in VQA-CP v2. We leave it for future work.

asing SoTAs with 1.55% and 5.79%, respectively, with fewer or similar amounts of parameters, establishing new state-of-the-arts. 2) Our methods (30% lxmert and 50% lxmert) outperform the debiased full lxmert, even with much fewer parameters. 3) Full lxmert(lpf) and full lxmert(lmh) are good at different question types, which can partly explain why combining them in different stages produces more robust subnetworks.

We also add experiments on a more recent VLP mPLUG (Li et al., 2022). We adopt the base version of mPLUG, fine-tune it on the VQA-CP v2 training set and then conduct pruning using mask training. Since mPLUG formulas VQA as a text generation task, we adopt the LPF debiasing method. Note that LMH and RUBi cannot be directly applied to debias text generation models, because they are designed for classification loss over a fixed number of classes. As shown in the bottom rows of Tab. 1, the mPLUG trained with standard cross-entropy (CE) loss can be simultaneously compressed (to 50%) and debiased (+5.48 Acc). The mPLUG trained with LPF debiasing loss can also be compressed to 50% with a slight accuracy decline. These results demonstrate that the findings and techniques present in our work can be generalized to more advanced VLPs.

Results on VQA-VS are presented in Tab. 2. We can observe that: 1) Our methods "bce+bce" 10% lxmert and "lmh+lmh" 30% lxmert outperform all the non-VLP debiasing methods in both ID and OOD settings, with similar or fewer parameters. 2) Except for LMH, other debiasing methods underperform BCE in OOD-mean. LMH improves the OOD accuracy at the cost of ID accuracy decline. 3) The "lmh+lmh" subnetworks

(even with 50% remaining parameters) obviously improve the ID performance of lxmert (lmh) and retain comparable OOD performance. 4) Compared with "bce+bce", the OOD advantage of "lmh+lmh" outweighs its ID disadvantage at 50% to 90% parameters. With fewer remaining parameters, the overall performance of "bce+bce" is superior.

## 4 Conclusion

To facilitate the application of VLP-based VQA systems, this paper presents the first joint study on the compression and debiasing problems of VLP for the VQA task. Through extensive experiments with three VLPs (i.e., lxmert, visualBERT and mPLUG), we analyze the impact of compression on the OOD generalization ability. We present a comprehensive study on the design of the training and compression pipeline for a good sparsity-performance trade-off, and provide some valuable findings about the assignment of sparsity to different modality-specific modules. The compressed lxmert subnetworks in this paper outperform the SoTA debiasing methods with fewer or similar model parameter counts.

## Limitations

Although we have empirically verified that the *adoption of modality-specific sparsity* is beneficial for the search for more robust subnetworks, our work still does not provide a solution on how to determine the optimal sparsity assignment effectively and efficiently. We invite follow-up studies to further address it in future work.

## Acknowledgement

This work was supported by National Natural Science Foundation of China (No. 61976207) and National Social Science Foundation of China (No. 21AZD145).

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

# A More Related Work

## A.1 Overcoming Dataset Bias in VQA

Most VQA systems heavily rely on the information of the question to predict answers no matter the content of the given image. That is they learned the language biases in datasets. They are not robust and always perform poor in the OOD setting where the language biases they learned in training set are invalid for test set. To promote the development of models that overcome such problem, VQA-CP v2 (Goyal et al., 2017) is proposed and has become the standard OOD benchmark in VQA. Currently, the widely used debiasing methods can be roughly grouped into non-data-augmentation (Clark et al., 2019; Liang et al., 2021b; Mahabadi and Henderson, 2019) and data-augmentation methods (Chen et al., 2020a; Gokhale et al., 2020). The former applies a **biased model** (trained with question only) to

regularize the model training and thus prevent learning from question. The latter generates samples to balance the training data and directly erase the biases in the training set. However, the augmented data also increase the training cost, and overcoming the language-bias problem remaining the original dataset biases unchanged still remains a major challenge (Liang et al., 2021b; Niu et al., 2021). Thus, we only focus on non-data-augmentation methods, such as LMH (Clark et al., 2019), RUBi (Cadene et al., 2019) and LPF (Liang et al., 2021b). Very recently, VQA-VS[6] (Si et al., 2022b) is proposed to explore the varying types of dataset biases. We also use this dataset to study how the training and compression pipeline affect different dataset biases.

## A.2 Vision-Language Pre-trained Models

Recently, VLPs (Dou et al., 2022; Li et al., 2021, 2020a; Wang et al., 2021a,b; Zhang et al., 2021; Si et al., 2023; Li et al., 2022) based on the Transformer backbone (Vaswani et al., 2017) have achieved encouraging success. Specially, OFA (Wang et al., 2022) and Florence (Yuan et al., 2021) establish the SoTA on the in-distribution VQA v2. To learn better cross-modality representations and vision-language alignment, they are trained with large-scale pre-training data and generally have huge model capacity. Among them, lxmert (Tan and Bansal, 2019) is the most widely used VLP as the backbone model in VQA field (e.g., some data-augmentation debiasing methods (Gokhale et al., 2020; Si et al., 2021; Wen et al., 2021) and the open-domain VQA (Marino et al., 2019) method MuKEA (Ding et al., 2022)). In this paper, we therefore mainly use lxmert as the backbone model and extend several debiasing methods to it for in-depth research on compressing and debiasing. For completeness, we also conduct experiments on the popular VLP visualBERT (Li et al., 2019).

## A.3 Model Compression and Robustness

Model compression techniques for Transformer-based pre-trained models are well developed (mainly around BERT), including pruning (Gale et al., 2019; Gordon et al., 2020; Michel et al., 2019), knowledge distillation (Jiao et al., 2020; Sanh et al., 2019; Sun et al., 2019), parameter sharing (Lan et al., 2020) and quantization (Zafrir et al., 2019; Zhang et al., 2020). Inspired by lottery ticket

hypothesis (Frankle and Carbin, 2019), many recent studies show that BERT can be pruned to a sparse subnetwork after (Gale et al., 2019) and before fine-tuning (Chen et al., 2020b; Liang et al., 2021a; Liu et al., 2022; Prasanna et al., 2020), without performance degrading. On this basis, we extend the pruning paradigm to the fine-tuned lxmert for OOD scenario in VQA, which incorporates the debiasing methods when fine-tuning and pruning. In the NLP and CV fields, some recent efforts have also been made to study model compression and robustness to adversarial attacks (Fu et al., 2021; Gui et al., 2019; Sehwag et al., 2020; Xu et al., 2021; Ye et al., 2019) and spurious correlations (Du et al., 2021; Xu et al., 2021) (which is more common than the worst-case adversarial attack). Dataset-bias problem is a typical symptom of spurious correlations and poses a challenge to VQA models. We are the first to thoroughly investigate the sparsity and OOD robustness for VLPs in VQA.

## B  More Details of Model and Implementation

### B.1  lxmert Architecture and Subnetworks

For lxmert, the embedding layer and visual fc layer map language-modality input (token sequences obtained by WordPiece tokenizer) and vision-modality input (36 object features obtained by Faster R-CNN (Ren et al., 2015)) into the same-dimension space. The pooler layer connects the Transformer top layer and the classifier. The Transformer layers involve three encoders [7]: language encoder ($L_{enc}$), object relationship encoder ($R_{enc}$) and cross-modality encoder ($C_{enc}$), and are usually composed of attention modules and feed-forward networks (FFN).

The attention modules have four kinds of weight matrices, i.e., the query, key and value matrices $\mathbf{W}_{Q,K,V} \in \mathbb{R}^{d_{model} \times d_{model}}$, and the output matrix $\mathbf{W}_O \in \mathbb{R}^{d_{model} \times d_{model}}$. FFN contains two linear layers $\mathbf{W}_{in} \in \mathbb{R}^{d_{model} \times d_{FFN}}$, $\mathbf{W}_{out} \in \mathbb{R}^{d_{FFN} \times d_{model}}$.

We adopt unstructured pruning to obtain a compressed version (i.e., a subnetwork) of the original VLPs. Specifically, given a VLP $f(\boldsymbol{\theta})$ with

---

[6]Both VQA-VS and VQA-CP v2 datasets are licensed under *Commons Attribution 4.0 International License.*

[7]Each Transformer layer of the language encoder and object relationship encoder has a multi-head self-attention module and a feed-forward network (FFN). Each Transformer layer of the cross-modality encoder has a language self-attention module, a visual self-attention module and a multi-head cross-attention module. Only the language self-attention and visual self-attention modules are followed by FFN. All the weight matrices of Transformer layers are summarized in eq. 12.

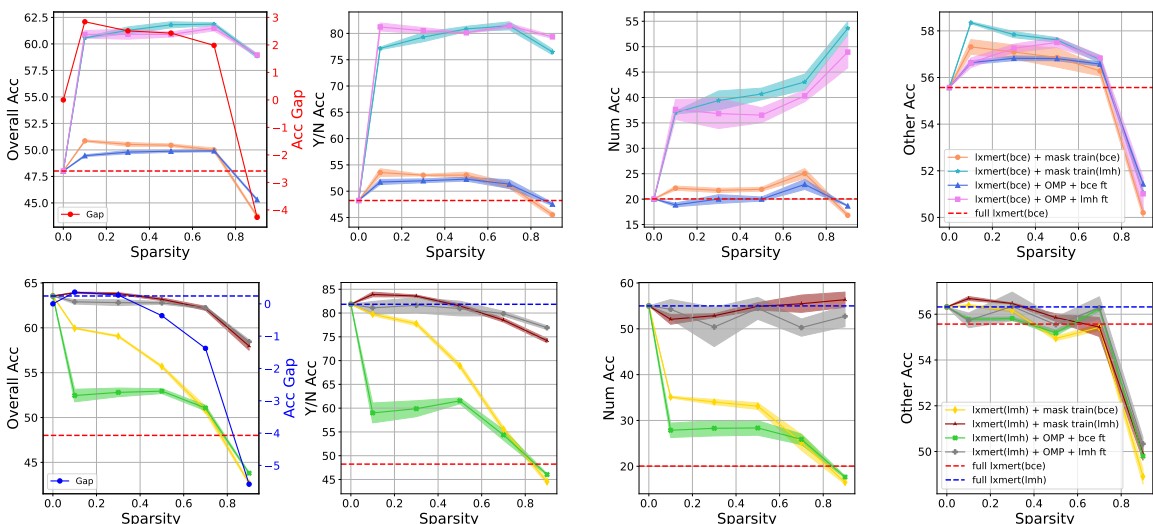

Figure 12: Results of subnetworks from the BCE fine-tuned lxmert (upper) and from the LMH fine-tuned lxmert (lower) on VQA-CP v2. "lxmert(bce/lmh)" denotes full model fine-tuning in Stage1, "mask train(bce/lmh)" and "OMP" denote pruning in Stage2. "bce/lmh ft" denotes further fine-tuning in Stage3. "Gap" denotes the improvement of mask train(bce/lmh) over full lxmert(bce/lmh).

parameters $\boldsymbol{\theta}$, we apply a binary pruning mask $\mathbf{m} \in \{0,1\}^{|\boldsymbol{\theta}|}$ to the model parameters, which gives rise to $f(\mathbf{m} \odot \boldsymbol{\theta})$, where $\odot$ is the element-wise product. For lxmert, we focus on the embedding layer, visual fc layer, pooler layer and Transformer layers of which the parameters are pre-trained, while the classifier is excluded. The language encoder, visual encoder, cross-modality encoder have $T$, $I$ and $X$ Transformer layers respectively. The parameters to be pruned are:

$$\boldsymbol{\theta}_{pr} = \{\mathbf{W}_{\text{emb}}, \mathbf{W}_{\text{vis-fc}}, \mathbf{W}_{\text{plr}}\} \cup \boldsymbol{\theta}_{L_{enc}} \cup \boldsymbol{\theta}_{R_{enc}} \cup \boldsymbol{\theta}_{X_{enc}} \tag{11}$$

where $\mathbf{W}_{\text{emb}}$, $\mathbf{W}_{\text{vis-fc}}$ and $\mathbf{W}_{\text{plr}}$ are the weights of embedding layer, vision fc layer and pool layer, $\boldsymbol{\theta}_{L_{enc}} \cup \boldsymbol{\theta}_{R_{enc}} \cup \boldsymbol{\theta}_{X_{enc}}$ are the parameters of Transformer layers:

$$\boldsymbol{\theta}_{L_{enc}} = \{\mathbf{W}_{Q_L}^t, \mathbf{W}_{K_L}^t, \mathbf{W}_{V_L}^t, \mathbf{W}_{O_L}^t, \mathbf{W}_{\text{in}_L}^t, \mathbf{W}_{\text{out}_L}^t\}_{t=1}^T$$
$$\boldsymbol{\theta}_{R_{enc}} = \{\mathbf{W}_{Q_R}^i, \mathbf{W}_{V_R}^i, \mathbf{W}_{K_R}^i, \mathbf{W}_{O_R}^i, \mathbf{W}_{\text{in}_R}^i, \mathbf{W}_{\text{out}_R}^i\}_{i=1}^I$$
$$\boldsymbol{\theta}_{X_{enc}} = \{\mathbf{W}_{Q_{CX}}^x, \mathbf{W}_{K_{CX}}^x, \mathbf{W}_{K_{CX}}^x, \mathbf{W}_{O_{CX}}^x,$$
$$\mathbf{W}_{Q_{CL}}^x, \mathbf{W}_{K_{CL}}^x, \mathbf{W}_{V_{CL}}^x, \mathbf{W}_{O_{CL}}^x, \mathbf{W}_{\text{in}_{CL}}^x, \mathbf{W}_{\text{out}_{CL}}^x,$$
$$\mathbf{W}_{Q_{CR}}^x, \mathbf{W}_{K_{CR}}^x, \mathbf{W}_{K_{CR}}^x, \mathbf{W}_{O_{CR}}^x, \mathbf{W}_{\text{in}_{CR}}^x, \mathbf{W}_{\text{out}_{CR}}^x\}_{x=1}^X \tag{12}$$

where $CX$, $CL$ and $CR$ are the language self-attention, visual self-attention and cross-attention modules respectively.

### B.2 visualBERT Architecture and Subnetworks

Similar to lxmert, visualBERT is composed of an embedding layer, a visual projection layer, a pooler layer, a stack of Transformer layers. Differently, visualBERT's Transformer layers only involve a single encoder ($V_{enc}$). The parameters of visualBERT to be pruned are:

$$\boldsymbol{\theta}_{pr} = \{\mathbf{W}_{\text{emb}}, \mathbf{W}_{\text{plr}}\} \cup \boldsymbol{\theta}_{V_{enc}} \tag{13}$$

where $\mathbf{W}_{\text{emb}}$ and $\mathbf{W}_{\text{plr}}$ are the weights of embedding layer and pool layer, $\boldsymbol{\theta}_{V_{enc}}$ are the parameters of Transformer layers:

$$\boldsymbol{\theta}_{V_{enc}} = \{\mathbf{W}_{Q_L}^v, \mathbf{W}_{K_L}^v, \mathbf{W}_{V_L}^v, \mathbf{W}_{O_L}^v, \mathbf{W}_{\text{in}_L}^v, \mathbf{W}_{\text{out}_L}^v\}_{v=1}^V \tag{14}$$

where $V = 12$.

### B.3 LMH details

LMH takes a step further based on Produce of Experts (PoE) (Hinton, 2002), which simply combines the predicted distributions of the main model and the biasd model as follows:

$$\hat{\mathbf{p}}_{deb} = softmax(log(\mathbf{p}_m) + log(\mathbf{p}_b)) \tag{15}$$

where $\mathbf{p}_b$ is the predicted distribution of biased model, and indicates the bias degree of the sample. In this way, when a sample is heavily biased, that is, $\mathbf{p}_b$ is large, the main model will not output a large $\mathbf{p}_m$ for it during training. Following (Clark et al., 2019), we directly use the answers' frequency under each question type as $\mathbf{p}_b$.

To selectively adjust the main model's behavior, LMH adds a learn function $g$ to explicitly deter-

mine how much to trust the learned biases:

$$\hat{\mathbf{p}}_{deb} = softmax(log(\mathbf{p}_m) + g(h)log(\mathbf{p}_b))$$
$$g(h) = softplus(w \cdot h) \tag{16}$$

where $h$ is the cross-modality representation from the last hidden layer of lxmert, $w$ is trainable. To prevent $\mathbf{p}_b$ being ignored, LMH also adds an entropy penalty item $R$, and the final loss is computed as:

$$\mathcal{L}_{lmh} = t \cdot log(\delta(\hat{\mathbf{p}}_{deb})) + (1 - t) \cdot log(1 - \delta(\hat{\mathbf{p}}_{deb}))] + R \tag{17}$$

### B.4 Model and Implementation Details

Lxmert has about 202M parameters, and 197.7M parameters are involved in the pruning process (4.5M parameters are left to the classifier). The three modules from different modalities, namely the language module, the visual module and the cross-modality module, contain 83.1M, 35.3M and 78.8M parameters respectively. We train the models for 20 epochs with a batch size of 128 on two Tesla-V100-32G or 256 on A100-80GB. The AdamW (Loshchilov and Hutter, 2017) optimizer is adopted with a learning rate of 5e-5. Our codes are based on the huggingface transformers library (Wolf et al., 2020). We adopt visualBERT of its *coco-pre* version which is pre-trained with COCO (Chen et al., 2015) dataset.

## C More Experiments on VQA-CP v2

### C.1 Performance of Subnetworks on Three Types of Questions

**Subnetworks from BCE Fine-tuned lxmert.** For the three types of questions, as shown in the right three plots of Fig. 12 (upper), we find that: 1) The performance on "Num" questions is sensitive to the varying sparsity levels while that on "Y/N" questions is relatively stable in general except at 90% sparsity. Specially, with the increase of sparsity, the performance on "Num" questions of "mask train(lmh)" and "OMP + lmh ft" counterintuitively greatly promote. This shows that language biases for the "Num" questions exist in a large proportion of the parameters of biased lxmert. 2) For the "Other" questions, debiasing methods have little gain on the performance of subnetworks. For example, the performance of "mask train(lmh)" is similar with that of "mask train(bce)". This indicates that the language biases for "Other" questions is minor

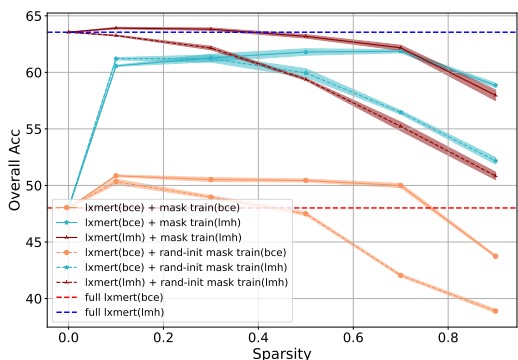

Figure 13: Results of subnetworks obtained by mask training with different initialization strategies of $\hat{\mathbf{m}}$ on VQA-CP v2. "rand-init" means initializing $\hat{\mathbf{m}}$ randomly.

in training set. Therefore, "Other" questions request more reasoning than debiasing. 3)There is a sharp decline of all the subnetworks' performance on "Other" questions from 70% $\sim$ 90% sparsity. We conjecture that this is because reducing the model's capacity too drastically hurt the reasoning ability which is necessary to answer the "Other" questions correctly.

**Subnetworks from LMH Fine-tuned lxmert.** The right three plots of Fig. 12 (lower) shows the performance of LMH fine-tuned lxmert subnetworks on different types of questions. For the "Num" questions, when compressing LMH fine-tuned lxmert (the grey and maroon lines), the performance of subnetworks no longer rises with sparsity growth. This demonstrates that language biases for the "Num" questions exist in a much smaller proportion of the parameters of debiased lxmert than that of biased lxmert. For "Other" questions, "lxmert(bce) + mask train(lmh)" is consistently superior to "lxmert(lmh) + mask train(lmh)", which demonstrates that further debiasing the debiased full lxmert in the pruning process sacrifices the reasoning ability.

### C.2 The Effect of Different Initialization Strategies of $\hat{\mathbf{m}}$ for Mask Training

We conduct experiments with different subnetworks to validate the effectiveness of initializing $\hat{\mathbf{m}}$ according to the magnitudes of lxmert's pretrained weights. From Fig. 13, it can be seen that "lxmert(bce) + mask train(bce)", "lxmert(bce) + mask train(lmh)", "lxmert(lmh) + mask train(bce)" (dashed lines) consistently outperform "lxmert(bce) + rand-init mask train(bce)", "lxmert(bce) + rand-init mask train(lmh)", "lxmert(lmh) + rand-init

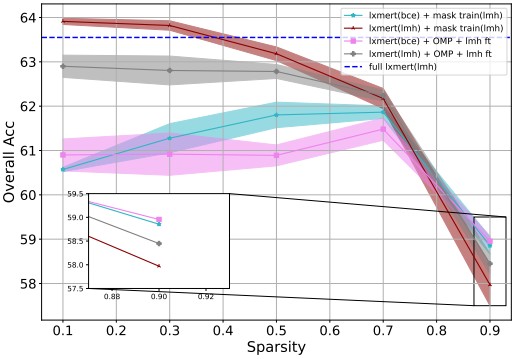

Figure 14: Results of subnetworks obtained by pruning with debiasing method LMH on VQA-CP v2.

mask train(bce)" (full lines) at all sparsity levels. As the sparsity increases, the gaps widen. This shows the initialization strategy we adopt is more effective than random initialization.

### C.3 A Close Look at The Performance of Subnetworks at 90% Sparsity

From Fig. 14, we derive two abnormal observations at the extremely high sparsity, i.e., 90%: 1) Pruning with "OMP + lmh ft" (pink and grey lines) is better than pruning with "mask train(lmh)" (cyan and brown lines). 2) Starting from "lxmert(bce)" (pink and cyan lines) is better than starting from "lxmert(lmh)" (grey and brown lines). The two observations at 90% sparsity are contrary to other sparsity. For the first observation, we conjecture that this is because mask training (which involves binarization and gradient estimation) is more difficult to optimize at 90% compared with further fine-tuning of the OMP subnetworks. The second observation can be explained by that: Further debiasing the debiased full lxmert in the pruning process slightly sacrifices the performance on "Other" questions, which require more reasoning ability than debiasing ability (as shown in the rightmost two plots of Fig. 12). Therefore, at the extremely high sparsity, when the benefits of debiasing on "Y/N" and "Num" questions are small, the performance penalty on "Other" questions results in a drop in "Overall" accuracy. Nevertheless, the gaps between "lxmert(lmh) + mask train(lmh)" and the other two pipelines are small at 90% sparsity.

### C.4 Sparsity Configurations for the Three Modality-specific Modules

For the overall target sparsity of 50% and 70%, we adopt the following procedure to search the comfortable zone for the modality-specific sparsity:

**First**, we traverse $[10\%, 30\%, 50\%, 70\%, 90\%]$ (i.e., step size of 20%) to assign modality-specific sparsity for any two modules, and compute the modality-specific sparsity for the remaining one[8] according to eq. 10 in the main paper. From the experimental results of these sparsity configurations, we can determine the approximate range where the pruned subnetworks perform better.

**Second**, we use the same method to traverse the reduced range with a smaller step size of 5%. In this way, we can determine the most comfortable zone for the modality-specific sparsity.

Similarly, when the overall target sparsity is 90%, we directly traverse $80\% \sim 98\%$ with a step size of 2% to search the most comfortable zone of the modality-specific sparsity.

## D More Experiments on VQA-VS

### D.1 Performance on varying OOD test sets of VQA-VS

**The Effect of Compression without Debiasing** For simplicity, we categorize the nine OOD test sets into 3 categories of different modalities, i.e., language-based (OOD-lang), visual-based (OOD-vis) and cross-modality (OOD-crsM) ones. We report the average accuracy of each category, as well as the IID accuracy and the average accuracy of all OOD test sets (OOD-mean) in Fig. 15.

The upper part of Fig. 15 shows the performance of subnetworks compressed without debiasing method, it can be seen that: 1) All subnetworks obtained by pruning all three modules underperform "full model(bce)" in ID test set. This is because the ID performance relies on memory ability, which is positively related to the parameter quantity. 2) The subnetworks obtained by pruning the language module consistently outperform the full model on OOD-mean, OOD-lang and OOD-crsM test sets, which are related to the language bias. This indicates that the language module of lxmert is slightly overparameterized. 3) In contrast, pruning other modules causes a negative impact on OOD performance. Especially, pruning visual modules also results in a sharp OOD-vis accuracy drop, indicating that the visual module of lxmert is not suitable for compression.

**The Effect of Compression with Debiasing** The lower part of Fig. 15 shows the VQA-VS perfor-

---

[8]We exclude the configurations where the computed sparsity for the remaining module is greater than 1 or smaller than 0.

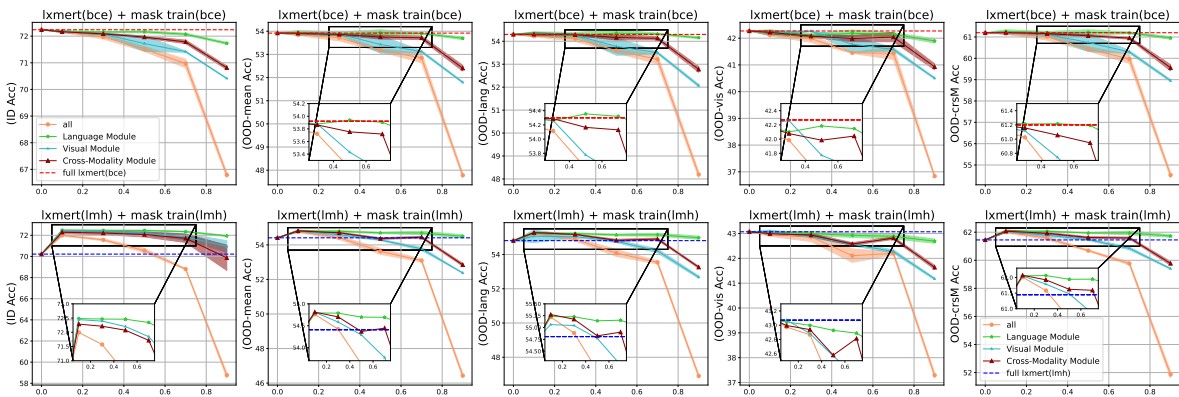

Figure 15: Results of subnetworks pruned using BCE (upper) and LMH (lower) on VQA-VS. Each column measures accuracy on ID test set, all, language-based, visual-based and cross-modality OOD test sets respectively. Different lines denote subnetworks obtained by pruning all, language, visual and cross-modality modules respectively.

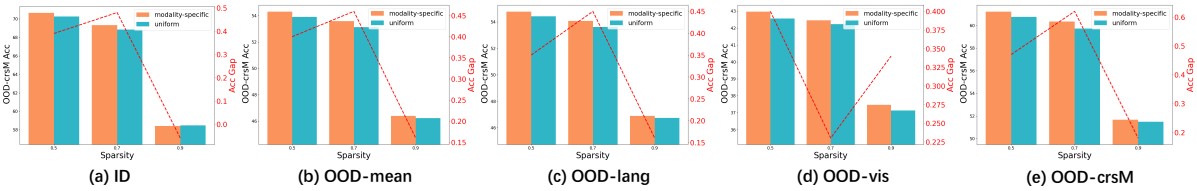

Figure 16: Comparison of subnetworks "lxmert(lmh) + mask train(lmh)" with uniform sparsity and modality-specific sparsity on VQA-VS.

mance of "lxmert(lmh) + mask train(lmh)"[9], which performs the best on VQA-CP v2. We can observe that: 1) Pruning any modules can improve ID performance over the debiased full model ("full model(lmh)"). This is because debiasing methods improve OOD performance at the cost of ID performance, while our pipeline alleviates such ID performance decline by compressing some harmful parameters. 2) Similarly, pruning any lxmert modules with a small sparsity (e.g., 0.2 and 0.4) also improves the OOD-mean performance. This demonstrates the existence of sparse and robust lxmert subnetworks on VQA-VS. 3) Especially, subnetworks obtained by compressing the language module consistently perform better than subnetworks obtained by pruning other modules and the debiased full model (except on OOD-vis), since the dataset biases tend to be learned by the language module. 4) However, pruning on any module fails to improve the OOD-vis accuracy, as the debiasing method LMH is designed for the language bias.

## D.2 The Effect of Modality-specific Sparsity on varying OOD test sets of VQA-VS

We directly use the modality-specific sparsity selected by the experiments of Sec. 3.4 in the main paper on VQA-CP v2. Fig. 16 shows that the subnetworks with modality-specific sparsity always outperform those with uniform sparsity except for 90% sparsity on ID test set, which validates that different modules should be compressed with different sparsity. Besides, when the overall sparsity is too large or too small, the benefits of the assignment of modality-specific sparsity will decrease accordingly. Note that the phenomenon of OOD-vis is different from other OOD test sets as the sparsity increases, since the debiasing methods LMH is designed for the language biases.

[9]Note that most debiasing methods fail on VQA-VS (Si et al., 2022b), such as LPF and RUBi. We thus do not discuss them in this section.
