# OpenReview forum: "Compressing and Debiasing Vision-Language Pre-Trained Models for Visual Question Answering"
_EMNLP/2023/Conference — EMNLP 2023 Main_

### Official Review · Reviewer_rJ51 · 2023-07-30

**Soundness:** 3

**Excitement:**

3: Ambivalent: It has merits (e.g., it reports state-of-the-art results, the idea is nice), but there are key weaknesses (e.g., it describes incremental work), and it can significantly benefit from another round of revision. However, I won't object to accepting it if my co-reviewers champion it.

**Paper Topic And Main Contributions:**

This paper presents the first joint study on compressing and debiasing vision-language pre-trained models (VLPs) for visual question answering (VQA). Although VLPs achieve good performance on standard VQA, they rely on dataset biases which hurts out-of-distribution (OOD) generalization, and are inefficient in terms of memory and computation. This paper investigates whether a VLP can be simultaneously compressed to reduce costs and debiased to improve OOD accuracy by searching for sparse and robust subnetworks. Through systematic analysis of training/compression pipeline designs and sparsity assignment to different modality modules, experiments with 2 VLPs, 2 compression methods, 4 training methods, 2 datasets and varying sparsity levels show it's possible to obtain sparse VLP subnetworks that match debiased full VLPs and outperform debiasing state-of-the-art on OOD datasets with fewer parameters.

**Reasons To Accept:**

1. This paper addresses two important challenges with vision-language pre-trained models (VLPs) - reliance on biases that hurts generalization, and inefficiency in terms of computation/memory.
2. The systematic analysis of training pipeline, compression techniques, and sparsity assignment provides new insights into effectively compressing and debiasing VLPs.
3. The authors propose joint compression-debiasing approach that obtained compressed VLP subnetworks that match full debiased models and outperform prior debiasing techniques. It provides a promising new technique for model compression and debiasing that could transfer to other NLP domains.



**Reasons To Reject:**

My major concern is that in equation (9), when trying to obtain the compressed model, the authors conducted out-of-distribution (OOD) evaluation. This means the authors utilized OOD data during the model development stage. This setting diverges from existing OOD generalization literature, where models are typically developed without access to OOD data. Since the authors have incorporated OOD data into model development, it is unsurprising they can simultaneously find compressed and robust models. The standard approach is to develop models only on in-distribution data, then evaluate robustness separately on held-out OOD data. Using OOD data during development confounds the results and limits conclusions about true OOD generalization abilities.

**Reproducibility:**

4: Could mostly reproduce the results, but there may be some variation because of sample variance or minor variations in their interpretation of the protocol or method.

**Reviewer Confidence:**

4: Quite sure. I tried to check the important points carefully. It's unlikely, though conceivable, that I missed something that should affect my ratings.

---

> ### Author Rebuttal · Authors · 2023-08-27
>
> +**Q1**: The concern of Equation9 & The data used in model development.
> > This is a misunderstanding. We would like to clarify three points:
> > (1) Equation 9 aims to describe our ultimate objective, rather than implying that "we utilize OOD data during the model development stage". We will revise this paragraph to avoid ambiguity.
> > (2) All of our experiments followed the same settings with all previous baselines on VQA-CP and VQA-VS to ensure comparative fairness.
> (3) We fully agree with your viewpoint that "models are typically developed without access to OOD data",  and have been implementing it as much as possible in this paper. However, due to the lack of a in-distribution validation set for VQA-CP ([1,2] mentioned this issue), we can only follow all previous debiasing methods's settings and directly use the test set to select checkpoints, because  VQA-CP is the most popular OOD benchmark in VQA, and almost all debiasing methods for VQA evaluates their models on this benchmark. Note that even in this setting, finding robust subnetworks is non-trivial, because the full model and subnetworks without debiasing training are not robust.
> However, to avoid the impact caused by this issue, we further experiment on the VQA-VS dataset. VQA-VS ensures that no OOD data is available during training and has addressed the limitations of VQA-CP from multiple aspects, setting a more reasonable standard for the OOD testing procedure. Our method can still find compressed and robust models on VQA-VS.
>
> [1] On the Value of Out-of-Distribution Testing: An Example of Goodhart's Law.
> [2] Language Prior Is Not the Only Shortcut: A Benchmark for Shortcut Learning in VQA.

---

### Official Review · Reviewer_AWxQ · 2023-08-03

**Soundness:** 3

**Excitement:**

3: Ambivalent: It has merits (e.g., it reports state-of-the-art results, the idea is nice), but there are key weaknesses (e.g., it describes incremental work), and it can significantly benefit from another round of revision. However, I won't object to accepting it if my co-reviewers champion it.

**Paper Topic And Main Contributions:**

This paper presents an empirical study outlining a procedure to address the twofold issue of a model's failure to generalize to out-of-distribution data and model compression challenges. The core contribution lies in the author's exploration of the integration of debiasing techniques, training strategies, and compression methods. Their insights could inspire on how compression impacts generalization, performance, and different modules within the model. The proposed subnetworks can achieve promising performance n the debiasing task.

**Questions For The Authors:**

What is the definition of a_k in the paper?

**Reasons To Accept:**

1. The comprehensive set of experiments in this paper effectively demonstrates that the sparsity introduced by compression can contribute to improved performance on out-of-distribution (OOD) tasks.

2. The proposed subnetwork can achieve promising results on two OOD datasets.

**Reasons To Reject:**

The technique contribution is limited because the used compression methods and debiasing methods are directly adopted from previous work.

Furthermore, it is not fair to compare the proposed subnetworks with other debiasing methods directly because the compression methods change the model itself, and the training strategies are searched as well, which limits the contribution of the proposed subnetworks.

**Reproducibility:**

3: Could reproduce the results with some difficulty. The settings of parameters are underspecified or subjectively determined; the training/evaluation data are not widely available.

**Reviewer Confidence:**

3: Pretty sure, but there's a chance I missed something. Although I have a good feel for this area in general, I did not carefully check the paper's details, e.g., the math, experimental design, or novelty.

---

> ### Author Rebuttal · Authors · 2023-08-27
>
> + **Q1**: Technical Contribution.
> >  We fully understand the concern of "technical novelty". However, we call for an inclusive mindset towards the contribution and novelty of our work, which is different from the regular papers that focus on individual techniques. We clarify our contribution from three perspectives:
> > **Firstly, we present the first joint study on debiasing and compression for the VQA task.** This topic is important as it can facilitate the deployment of SoTA VLPs in resource-constrained and security-sensitive real-world scenarios. It has been overlooked in existing literature, and our research fills this gap.
> >    **Secondly, from the empirical perspective,** our extensive experiments on the training and compression pipeline and sparsity assignment can serve as a valuable guideline for future research on this topic.
> >    **Thirdly, from the technical perspective,** we derive a specific compression and debiasing pipeline that establishes a SoTA sparsity-robustness trade-off. In contrast, neither existing debiasing techniques nor compression techniques alone can accomplish the objective of our topic.
>
> + **Q2**: Fairness of comparison.
> > We respectfully disagree. It is worth noting that the comparison of entire systems is common when benchmarking the SoTA methods, and the previous VQA debiasing SoTAs already employ different backbone models. We compare our debiased subnetworks with previous debiasing methods at a "system-level" to demonstrate a progress made in this field. Moreover, even when using the same backbone model, the subnetworks surpass the debiased lxmert (lmh) and lxmert (lpf) in terms of OOD performance, while also requiring significantly fewer model parameters.
>
> + **Q3**: What is the definition of $a_k$ in the paper?
> > Thank you for pointing out the missed details. $a_k$ denotes the ground-truth answer for the $k_{th}$ sample.

---

### Official Review · Reviewer_HrWM · 2023-08-08

**Soundness:** 4

**Excitement:**

4: Strong: This paper deepens the understanding of some phenomenon or lowers the barriers to an existing research direction.

**Paper Topic And Main Contributions:**

This research delves into compression techniques for the VLP model, specifically within the domain of Visual Question Answering (VQA). Using the LXMERT model as a baseline, the authors concurrently examine compression and debiasing issues. They explore various pruning strategies, including magnitude-based pruning and mask training, along with multiple debiasing approaches.
Experimental findings indicate that the compressed LXMERT subnetworks surpass state-of-the-art (SOTA) debiasing methods, achieving this performance with fewer or comparable model parameters.



**Reasons To Accept:**

1. The concept of simultaneously compressing and debiasing the VLP model presents an intriguing approach.
2. The proposed methodology demonstrates its efficacy when applied to the LXMERT model.

**Reasons To Reject:**

The authors restrict their experiments to the LXMERT model. It raises the question of whether similar performance enhancements would be observed with other VLP models.

**Reproducibility:**

4: Could mostly reproduce the results, but there may be some variation because of sample variance or minor variations in their interpretation of the protocol or method.

**Reviewer Confidence:**

2: Willing to defend my evaluation, but it is fairly likely that I missed some details, didn't understand some central points, or can't be sure about the novelty of the work.

---

> ### Author Rebuttal · Authors · 2023-08-27
>
> + **Q1**: Restrict the experiments to the LXMERT model?
> > **We did not only conduct experiments on LXMERT model, but also validated our method on visualBERT**.   We would like to clarify the reason of using LXMERT and VisualBERT in our experiments: Most reasearch on the VQA debiasing still use plain models (e.g., UpDn and S-MRL) [1,2,3]. Among the few VLPs used in this field, LXMERT and visualBERT are the most popular ones.  As we are the first joint study on VQA debiasing and model compressing, we choose these two representative VLPs to conduct thorough experiments. We believe our findings can generalize to varying transformer-based VLPs.
> > [1] https://paperswithcode.com/sota/visual-question-answering-on-vqa-cp
> > [2] https://github.com/cdancette/vqa-cp-leaderboard
> > [3] https://aclanthology.org/2022.findings-emnlp.495.pdf

---

### Official Review · Reviewer_KVsc · 2023-08-10

**Soundness:** 4

**Ethical Concerns:**

Yes

**Excitement:**

3: Ambivalent: It has merits (e.g., it reports state-of-the-art results, the idea is nice), but there are key weaknesses (e.g., it describes incremental work), and it can significantly benefit from another round of revision. However, I won't object to accepting it if my co-reviewers champion it.

**Paper Topic And Main Contributions:**

In this paper, the author  investigates whether a VLP can be compressed and debiased simultaneously by searching sparse and robust subnetworks.
They present a comprehensive study on the design of the training and compression pipeline for a good sparsity-performance trade-off, and provide some valuable findings about the assignment of sparsity to different modality-specific modules. The compressed lxmert subnetworks in this paper outperform the SoTA debiasing methods with fewer or similar model parameter counts.

**Questions For The Authors:**

see weakness.

**Reasons To Accept:**

1. The motivation of the paper is clear, and the approach is well-articulated.

2. The method proposed in this paper achieves state-of-the-art (SOTA) results on some datasets.

3. This paper investigates the relationship between sparsity and out-of-distribution (OOD) generalization performance.

**Reasons To Reject:**

1. The backbone models used in this paper, namely VisuaBERT and LXMERT, are relatively outdated. How does the method proposed in this paper perform on newer VLP (Vision-Language Pretraining) models?

2. This paper employs existing methods such as Magnitude-based Pruning in pruning, Mask Training, and LMH, RUBi, LPF in de-biasing, combining them into a single pipeline and applying them to the task proposed in this paper. I think the novelty of this paper is incremental in nature, and the authors need to further clarify their contributions.

3. There are some formatting issues and typographical errors in the paper, such as in Figure 2, where the last line of the legend, "full lxmbert," is missing a closing parenthesis.

4. The idea of using mask training and DiffPruning [1] seems very similar. Have the authors noticed the connection between the two?

5. The authors claim that "this paper presents the first joint study on the compression and debiasing problems of VLP for the VQA task." Has the paper compared the performance of previous methods in one domain with methods in another domain to demonstrate that the method proposed in this paper indeed makes a substantial contribution compared to previous methods? The authors need to reflect on this point.

[1] Demi Guo, Alexander M. Rush, and Yoon Kim. 2020. Parameter-efficient transfer learning with diff pruning. In Annual Meeting of the Association for Computational Linguistics.

**Reproducibility:**

3: Could reproduce the results with some difficulty. The settings of parameters are underspecified or subjectively determined; the training/evaluation data are not widely available.

**Reviewer Confidence:**

3: Pretty sure, but there's a chance I missed something. Although I have a good feel for this area in general, I did not carefully check the paper's details, e.g., the math, experimental design, or novelty.

---

> ### Author Rebuttal · Authors · 2023-08-27
>
> + **Q1**: Experiments on more recent VLPs.
> >   We can understand this concern. However, we would like to clarify the reason of using LXMERT and VisualBERT in our experiments: Most reasearch on the VQA debiasing still use plain models (e.g., UpDn and S-MRL) [1]. **Among the few VLPs used in this field, LXMERT and visualBERT are the most popular ones**[2,3].  As we are the first joint study on VQA debiasing and model compressing, we choose these two representative VLPs to conduct thorough experiments. We believe our findings can generalize to varying transformer-based VLPs.
>
> + **Q2**: Technical Contribution.
> >  We fully understand the concern of "technical novelty". However, _we call for an inclusive mindset towards the contribution and novelty of our work, which is different from the regular papers that focus on individual techniques_. We clarify our contribution from three perspectives:
> >   **Firstly, we present the first joint study on debiasing and compression** for the VQA task. This topic is important as it can facilitate the deployment of SoTA VLPs in resource-constrained and security-sensitive real-world scenarios. It has been overlooked in existing literature, and our research fills this gap.
> >   **Secondly, from the empirical perspective**, our extensive experiments on the training and compression pipeline and sparsity assignment can serve as a valuable guideline for future research on this topic.
> >    **Thirdly, from the technical perspective**, we derive a specific compression and debiasing pipeline that establishes a SoTA sparsity-robustness trade-off. In contrast, neither existing debiasing techniques nor compression techniques alone can accomplish the objective of our topic.
>
> + **Q3**: Formatting Issues.
> >  Thank you for the feedback. The questions and formatting issues will be properly addressed in the next version of our paper.
>
> + **Q4**: Connection between mask training and DiffPruning.
> >   Mask training and DiffPruning are similar in the training mechanism: they both optimize the mask vectors (referred to as Diff vector in DiffPruning) applied to the pre-trained parameters. However, they are fundamentally different in their objectives. DiffPruning aims at **reducing the newly added parameters** when transferring pre-trained models to a downstream task, while the model size is unchanged (the Diff vector is added to the frozen pre-trained parameters). In comparison, mask training aims at **obtaining a smaller model for inference** (a portion of parameters are set to zero based on the mask).
>
> + **Q5**: Has the paper compared the performance of previous methods in one domain with methods in another domain?
> > Yes, we have compared the methods in the debiasing domain (e.g., lxmert(lmh), lxmert(lpf), GCE, etc. in Fig.1) and methods the model compression domain (e.g., lxmert(bce)+mask train(bce) and lxmert(bce)+OMP(bce) in Fig.2). The debiased lxmert, while achieving strong OOD performance, is large in size and inefficient as the original lxmert. The compressed lxmert subnetworks (without debiasing), while more efficient, are only slightly better than original lxmert  in the OOD scenario. By combining the compression and debiasing techniques, we obtain lxmert subnetworks that are both small in size and achieve strong OOD performance, outperforming individual debiasing and compression methods in terms of the sparsity-robustness trade-off. We believe that this is a substantial contribution compared to previous methods.
>
>
>
> [1] https://paperswithcode.com/sota/visual-question-answering-on-vqa-cp
>
> [2] https://github.com/cdancette/vqa-cp-leaderboard
>
> [3] https://aclanthology.org/2022.findings-emnlp.495.pdf

---

### Meta-Review · Area_Chair_scfR · 2023-09-15

**Recommendation:** 5

**Metareview:**

I appreciate authors' full engagement and attempt to answer concerns and questions about the manuscript. The manuscript motivates the problem and provide insight into tradeoff between sparsity and performance and achieve performance parity with significant parameter reduction. While relation between smoothing, compression and bias reduction have been studied in traditional ML techniques, Authors have identified an opportunity to debias and compress at the same time for VLM, which in my view is novel and exciting and get facilitate further research in this area.

---

### Decision · Program_Chairs · 2023-10-07

**Decision:**

Accept-Main

**Comment:**

I appreciate authors' full engagement and attempt to answer concerns and questions about the manuscript. The manuscript motivates the problem and provide insight into tradeoff between sparsity and performance and achieve performance parity with significant parameter reduction. While relation between smoothing, compression and bias reduction have been studied in traditional ML techniques, Authors have identified an opportunity to debias and compress at the same time for VLM, which in my view is novel and exciting and get facilitate further research in this area.